# Radiosensitivity Differences between *EGFR* Mutant and Wild-Type Lung Cancer Cells are Larger at Lower Doses

**DOI:** 10.3390/ijms20153635

**Published:** 2019-07-25

**Authors:** Mai Anakura, Ankita Nachankar, Daijiro Kobayashi, Napapat Amornwichet, Yuka Hirota, Atsushi Shibata, Takahiro Oike, Takashi Nakano

**Affiliations:** 1Department of Radiation Oncology, Gunma University Graduate School of Medicine, Maebashi 371-8511, Japan; 2Division of Radiation Oncology, Department of Radiology, Faculty of Medicine, Chulalongkorn University, Bangkok 10330, Thailand; 3Gunma University Initiative for Advanced Research (GIAR), Gunma University, Maebashi 371-8511, Japan

**Keywords:** precision medicine, radiation therapy, radiosensitivity, clonogenic assays, gene mutations

## Abstract

In the era of precision medicine, radiotherapy strategies should be determined based on genetic profiles that predict tumor radiosensitivity. Accordingly, pre-clinical research aimed at discovering clinically applicable genetic profiles is needed. However, how a given genetic profile affects cancer cell radiosensitivity is unclear. To address this issue, we performed a pilot in vitro study by utilizing *EGFR* mutational status as a model for genetic profile. Clonogenic assays of *EGFR* mutant (*n* = 6) and wild-type (*n* = 9) non-small cell lung carcinoma (NSCLC) cell lines were performed independently by two oncologists. Clonogenic survival parameters SF_2_, SF_4_, SF_6_, SF_8_, mean inactivation dose (MID), D_10_, D_50_, α, and β were obtained using the linear quadratic model. The differences in the clonogenic survival parameters between the *EGFR* mutant and wild-type cell lines were assessed using the Mann–Whitney U test. As a result, for both datasets, the *p* values for SF_2_, SF_4_, D_50_, α, and α/β were below 0.05, and those for SF_2_ were lowest. These data indicate that a genetic profile of NSCLC cell lines might be predictive for their radiation response; i.e., *EGFR* mutant cell lines might be more sensitive to low dose- and low fraction sized-irradiation.

## 1. Introduction

Radiation therapy is one of the most important treatments for cancer. In the clinical practice of definitive radiation therapy, the doses required for clinical tumor control vary widely even for tumors arising in the same anatomical sites. For example, an integrated analysis of 50 single-institution studies performed by Okunieff et al. revealed that the dose required for 50% clinical tumor control ranged from 21.4 to 90.3 Gy for breast cancer, 50.4 to 83.4 Gy for supraglottic cancer, and 24.3 to 64.4 Gy for cervical cancer [1]. These data highlight the need to establish predictive biomarkers of tumor radioresponsiveness that could advance personalization of radiation therapy.

In recent years, the concept of precision medicine, i.e., stratification of treatment strategy based on individual patients’ genetic profiles, has become widespread in cancer treatment, in part due to technological advances in next-generation sequencing [2]. Drugs that target tumors carrying specific genetic profiles have yielded favorable outcomes in the clinic [3,4,5,6]. This indicates that a given genetic profile may affect cancer cell radiosensitivity as well.

Clonogenic assays are the gold standard for assessing cancer cell radiosensitivity in pre-clinical settings [7]. Evidence compiled over the past few decades suggests that the radiosensitivity of cancer cells, as determined by clonogenic assays, is relevant to tumor response to radiation therapy [8,9,10,11]. Multiple clonogenic survival parameters have been used as radiosensitivity endpoints for clonogenic assays [12]. However, how a given genetic profile affects cancer cell radiosensitivity as assessed by various clonogenic survival parameters is unclear. To address this issue, we performed a pilot study by utilizing *EGFR* mutational status as a model for genetic profile.

## 2. Results

To analyze the association of a given genetic profile with clonogenic survival parameters, we chose to analyze *EGFR* mutation status in non-small cell lung carcinoma (NSCLC); *EGFR* status is associated with the response of NSCLCs to radiation therapy [13]. Two independent oncologists performed clonogenic assays after X-ray irradiation using *EGFR* mutant (*n* = 6) and wild-type (*n* = 9) cell lines; hereafter, the two datasets obtained are referred to as datasets A and B (Figure 1 and Figure 2). The following clonogenic survival parameters were obtained: SF_2_, SF_4_, SF_6_, SF_8_, α, and β of the linear quadratic (LQ) model, D_10_, D_50_, and the mean inactivation dose (MID) (Table 1). Radiosensitivities for the individual cell lines showed significantly high correlation between dataset A and B in terms of all the clonogenic survival parameters (Table 2), indicating technical robustness of the experiments.

We then examined differences in the clonogenic survival parameters between the *EGFR* mutant and wild-type cell lines. In both datasets, the *p* values for SF_2_, SF_4_, D_50_, α, and α/β were below 0.05, and those for SF_2_ were the lowest (Table 1 and Figure 3). These data indicate that *EGFR* mutant cell lines might be more sensitive to low dose- and low fraction sized-irradiation.

There were no significant differences in SF_2_ between the cell lines carrying Δ746_750 (*n* = 3) and those carrying L858R (*n* = 3) for both datasets A and B (*p* = 0.20 and 0.70, respectively). This is reasonable in light of the clinical situation where these two mutations are used without functional distinction as indicators of EGFR-upregulated tumors targeted with its tyrosine kinase inhibitors.

## 3. Discussion

A large body of evidence supports the ability of *EGFR* status to predict tumor radioresponsiveness. In the pre-clinical setting, Amornwichet et al. [14] found that radiosensitivity determined by D_10_ from clonogenic assays was significantly higher for *EGFR* mutant cells than for *EGFR* wild-type cells in a panel of NSCLC cell lines, as well as in genetically engineered isogenic NSCLC cells. On the other hand, in a clinical setting, Yagishita et al. [13] retrospectively analyzed the outcomes of non-squamous NSCLC patients, and showed that the frequency of local relapse after chemo-radiation therapy was significantly lower in patients with *EGFR* mutations than in those with wild-type *EGFR* (4% versus 21%). Importantly, those two studies, as well as this study, analyzed the same activating mutations in *EGFR* (i.e., exon 19 in-frame deletion and exon 21 L858R missense mutation); therefore, the data in these studies are comparable. From the standpoint of mechanism, multiple groups have shown that mutations in *EGFR* cause defects in EGFR translocation to the nucleus and binding of EGFR to the catalytic subunit of DNA-dependent protein kinase (DNA-PKcs), a protein that plays pivotal roles in non-homologous end joining (NHEJ) of DNA double-strand breaks (DSBs) induced by ionizing irradiation [15,16,17]. Amornwichet et al. [14] showed that the increase in X-ray-induced γH2AX foci, a marker of DSBs, upon addition of NU7441, an inhibitor of NHEJ, is significantly smaller in *EGFR* mutant cells than in wild-type cells. These findings suggest that the high radiosensitivity of *EGFR* mutant cancer cells is at least in part based on reduced NHEJ activity associated with dysfunction of DNA-PKcs in response to ionizing irradiation. Together, these data support the scientific validity of our study design, which analyzed *EGFR* status as a model for a candidate genetic profile associated with cancer cell radiosensitivity as assessed by clonogenic assays.

In this study, the *p* values for the comparison between *EGFR* mutant and wild-type cell lines had the same relationship (SF_2_ < SF_4_ < SF_6_ < SF_8_) in datasets A and B. In addition, the differences in α values between the two groups reached statistical significance in both datasets, whereas the differences in β values did not. Thus, for assessment of cancer cell radiosensitivity using clonogenic assays, surviving fractions in the low dose range, where the linear component of the LQ model is dominant, better predict genetic profiles associated with clinical tumor radioresponsiveness. Over the past few decades, multiple groups have investigated the correlation between radiosensitivity, as determined by various clonogenic survival parameters, and the clinical responses of tumors to radiation therapy [12]. Fertil and Malaise [8] analyzed 59 survival curves derived from human cell lines, and found that SF_2_ was associated with the clinical dose required for tumor control. The same group analyzed an additional 101 survival curves derived from human cell lines, and found that radiosensitivity expressed in terms of SF_2_, α, and MID reflected the clinical responsiveness of the tumor from which a cell line was derived [11]. Daecon et al. analyzed the data on 51 human tumor cell lines, and found that the steepness of the initial slope of the survival curve is associated with the clinical response of a tumor to radiation therapy [9]. Although these previous studies stratified cells by histological type or primary tumor site, but not genetic profile, they share key findings with the present study. Based on these findings, we can conclude that the importance of parameters related to the initial slope of the survival curve has been re-confirmed in the context of precision medicine. From another aspect, NSCLC patients are nowadays often treated with hypofractionation schemes. Our data showing no radiosensitivity differences between *EGFR* mutant and wild-type groups at high doses (i.e., SF_8_ and SF_6_) suggest that hypofractionated irradiation employing high fraction size is likely to achieve comparable tumor control regardless of *EGFR* status.

Previous studies show that the mean SF_2_ differs among different cancer types (e.g., the mean SF_2_ is higher for glioblastoma cell line group compared to that for lymphoma cell line group). Nevertheless, SF_2_ show large variation among the cell lines within the same cancer type. From this perspective, SF_2_ should not be used as a clinical test to predict the response of an individual tumor to radiotherapy. On the other hand, evidence suggests clinical applicability of genetic profile-based algorithms generated from correlation analyses of in vitro SF_2_ data and the genetic profile counterparts [18,19,20,21]. Together, our data support the notion that SF_2_ is useful for pre-clinical study aiming at establishment of genetic profiles that predict tumor radioresponsiveness.

We previously analyzed the inter-study precision of clonogenic survival parameters in a given cell line [22]; in that study we found that SF_2_ and D_10_ have acceptable inter-study precision as a measure for radiosensitivity assessment. The data support the usefulness of SF_2_ from another aspect, i.e., technical robustness of the assay.

A limitation of this study is that we only analyzed the association with clonogenic survival parameters using a single gene (i.e., *EGFR*) because this study was a pilot attempt. Therefore, the results of this study cannot be applied directly to the clinic. Realistically, radiosensitivity of the tested cell lines must be influenced not only by *EGFR* but also by other genetic aberrations; therefore, clinical tumor radioresponsiveness will be more accurately predicted by genetic profiles comprising multiple genes, which warrants further investigation.

In summary, we showed that radiosensitivity differences between *EGFR* mutant and wild-type lung cancer cells are larger at lower doses. These findings may be useful for optimization of radiotherapy schemes to treat NSCLCs.

## 4. Materials and Methods

### 4.1. Cell Lines

A427, A549, H1299, H1650, H1703, H1975, H460, H520, H522, and HCC827 were obtained from ATCC (Manassas, VA, USA). II-18 and LK2 were obtained from the Japanese Cancer Resources Bank. H157 was provided by Dr. Harris (National Cancer Institute, Bethesda, Rockville, MD, USA). Ma-24 was provided by Dr. Shimizu (Tokushima University, Tokushima, Japan). PC9 was provided by Dr. Kato (Tokyo Medical College, Tokyo, Japan). The *EGFR* status of these cell lines is summarized in Table 3. All cell lines were cultured in RPMI-1640 (Sigma-Aldrich, St. Louis, MO, USA) supplemented with 10% fetal bovine serum (Life Technologies, Carlsbad, CA, USA).

### 4.2. Clonogenic Assays

For each of the 15 NSCLC cell lines, clonogenic survival parameters were obtained independently by two radiation oncologists (datasets A and B, respectively).

The experimental procedure was standardized among practitioners using the following protocol. Cells were passaged at least two times after thawing from freeze stocks, and the cells in log phase growth were used for the experiments. Cells were detached from culture dishes using trypsin (Sigma-Aldrich, St. Louis, MO, USA) and prepared as single-cell suspension in culture media. The cells were counted using a hemocytometer under an inverted microscope. The cell suspension was subjected to two serial dilutions at 1:10 (i.e., 1:100 dilution in total), from which the cell suspension containing the intended number of cells was prepared. The cells were seeded in 6-well plates; the number of seeded cells was determined based on the plating efficiency and the intrinsic radiosensitivities estimated by preparatory experiments (Table 4). After incubation for the minimum possible period of time for the seeded cells to attach on the plates (6–12 h, depending on cell lines), the cells were exposed to 2, 4, 6, or 8 Gy X-rays using a Faxitron RX-650 irradiator (100 kVp, 1.14 Gy/min; Faxitron Bioptics, Tucson, AZ, USA). After incubation for an additional 6–10 days, the cells were fixed with methanol and stained with crystal violet. Colonies comprising at least 50 cells were counted under an inverted microscope. The experiments were performed at least in triplicate, and the mean values for the surviving fractions were calculated after normalization to the corresponding un-irradiated controls. These survival data were fitted to the LQ model [29], and then α, β, D_10_, and D_50_ were calculated (Figure 4A). MID was calculated as previously described (Figure 4B) [10].

### 4.3. Statistical Analysis

Clonogenic survival parameters were compared between *EGFR* mutant and wild-type cell lines by non-parametric two-sided Mann–Whitney U test. Correlation of the survival data between datasets was examined by Spearman’s rank order test. Differences were considered statistically significant at *p* < 0.05. Analyses were performed using Prism8 (GraphPad, San Diego, CA, USA).

## Figures and Tables

**Figure 1 ijms-20-03635-f001:**
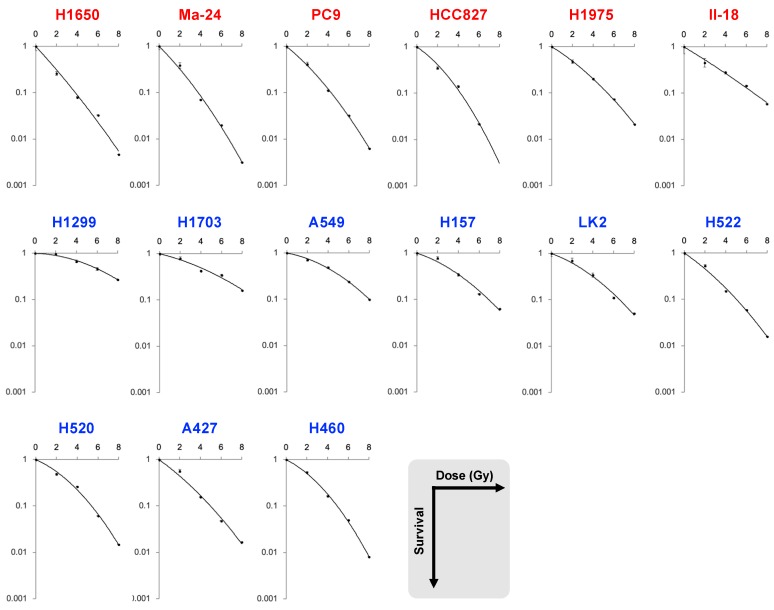
Dataset A: Survival curves for *EGFR* mutant (red) and wild-type (blue) non-small cell lung carcinoma cell lines treated with X-rays.

**Figure 2 ijms-20-03635-f002:**
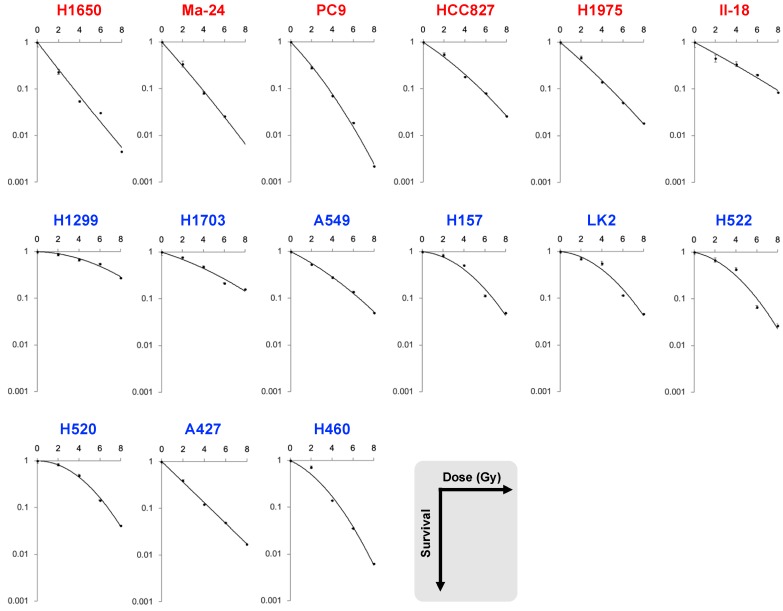
Dataset B: Survival curves for *EGFR* mutant (red) and wild-type (blue) non-small cell lung carcinoma cell lines treated with X-rays.

**Figure 3 ijms-20-03635-f003:**
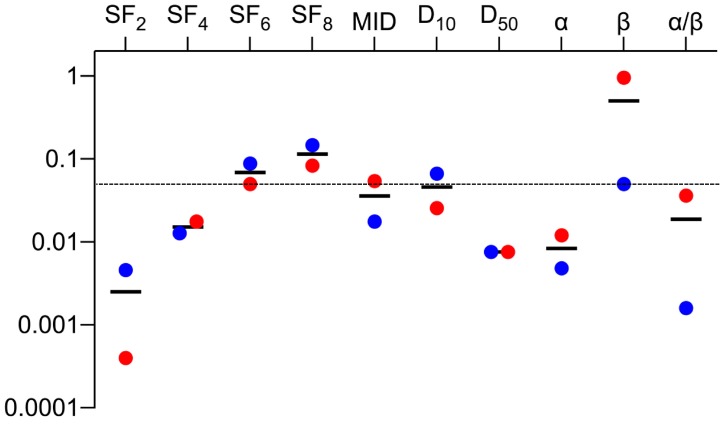
Summary of the *p* values for datasets A and B. MID, mean inactivation dose. Black lines indicate the mean values.

**Figure 4 ijms-20-03635-f004:**
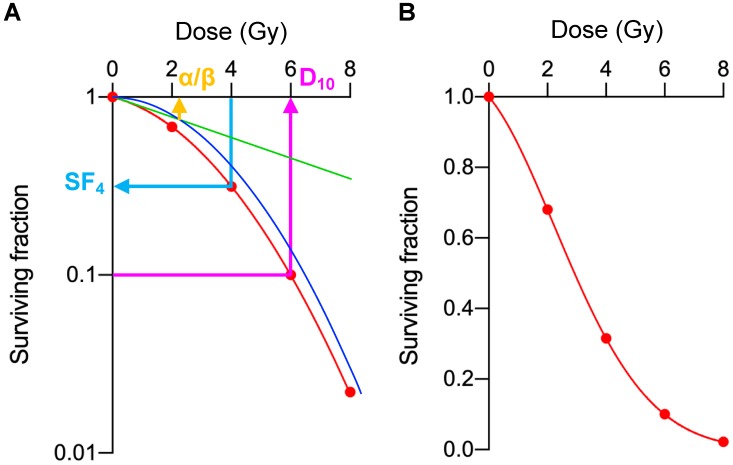
Exemplary presentation of the analyzed parameters for clonogenic survival. (**A**) Surviving fractions for the cells irradiated with 2, 4, 6, or 8 Gy (SF_2_, SF_4_, SF_6_, and SF_8_, respectively) are plotted on a semi-logarithmic-scaled graph (indicated as red dots). The survival data were fitted to the LQ model: *S* = exp(−(α*D* + β*D*^2^)), where *S* is the surviving fraction and *D* is the dose (red line indicates the LQ curve; green and blue line indicates its linear and quadratic component, respectively). D_10_ and D_50_ are calculated from the LQ model formula, where D_X_ indicates the dose that decreases the surviving fraction to X%. As an example, α/β (2.1), SF_4_ (0.32), and D_10_ (6.0) are shown as orange, light blue, and violet arrow, respectively. (**B**) The same exemplary survival data were plotted on a linear-scaled graph. MID equals the area under the curve (indicated as light red) [11].

**Table 1 ijms-20-03635-t001:** Clonogenic survival parameters for datasets A and B.

Dataset	Cell Line	*EGFR*	SF_2_	SF_4_	SF_6_	SF_8_	MID	D_10_	D_50_	α	β	α/β
A	H1650	mutant	0.26	0.079	0.033	0.0047	1.7	3.8	1.2	0.56	0.012	46.4
	Ma-24		0.39	0.070	0.020	0.0031	1.7	3.8	1.3	0.51	0.026	19.7
	PC9		0.42	0.11	0.032	0.0063	2.0	4.3	1.5	0.42	0.027	15.4
	HCC827		0.35	0.14	0.022	NA	2.0	4.2	1.6	0.34	0.048	7.2
	H1975		0.47	0.20	0.073	0.021	2.5	5.4	1.9	0.32	0.021	15.5
	II-18		0.45	0.28	0.15	0.058	3.0	6.7	2.2	0.31	0.0047	65.9
	H1299	wild-type	0.97	0.67	0.46	0.27	6.1	10.7	5.7	0.015	0.019	0.82
	H1703		0.79	0.43	0.35	0.16	4.7	9.5	4.0	0.13	0.012	10.3
	A549		0.71	0.49	0.24	0.098	4.2	8.0	3.8	0.092	0.025	3.7
	H157		0.78	0.34	0.13	0.062	3.4	6.9	2.9	0.17	0.024	7.2
	LK2		0.70	0.34	0.11	0.051	3.2	6.6	2.7	0.19	0.025	7.5
	H522		0.53	0.15	0.059	0.016	2.3	5.0	1.8	0.35	0.021	17.0
	H520		0.49	0.26	0.061	0.015	2.6	5.4	2.3	0.22	0.038	5.8
	A427		0.57	0.15	0.047	0.017	2.2	5.0	1.7	0.37	0.019	19.3
	H460		0.53	0.16	0.050	0.0081	2.4	4.9	2.0	0.26	0.043	6.0
	***p value***		***<0.001***	***0.018***	***0.049***	***0.082***	***0.054***	***0.025***	***0.0076***	***0.012***	***0.95***	***0.036***
B	H1650	mutant	0.23	0.054	0.031	0.0072	1.3	3.4	0.95	0.75	−0.018	NA
	Ma-24		0.34	0.081	0.026	NA	1.7	3.8	1.2	0.58	0.0068	84.4
	PC9		0.28	0.070	0.019	0.0022	1.6	3.6	1.2	0.53	0.028	19.2
	HCC827		0.55	0.18	0.081	0.026	2.5	5.5	1.9	0.34	0.015	22.8
	H1975		0.47	0.14	0.051	0.018	2.1	4.8	1.5	0.44	0.0082	53.8
	II-18		0.45	0.34	0.20	0.085	3.4	7.8	2.5	0.27	0.0029	93.3
	H1299	wild-type	0.87	0.67	0.55	0.28	6.3	11.1	5.9	0.015	0.017	0.85
	H1703		0.77	0.49	0.22	0.16	4.3	9.1	3.6	0.16	0.010	15.5
	A549		0.53	0.28	0.14	0.049	3.0	6.6	2.4	0.26	0.014	17.9
	H157		0.83	0.51	0.11	0.049	3.8	6.8	3.5	0.042	0.044	1.0
	LK2		0.70	0.56	0.12	0.046	3.8	6.7	3.5	0.044	0.044	1.0
	H522		0.68	0.43	0.066	0.027	3.2	6.0	2.9	0.11	0.046	2.3
	H520		0.83	0.49	0.14	0.041	3.9	6.8	3.7	−0.005	0.051	NA
	A427		0.39	0.12	0.049	0.021	1.9	4.5	1.3	0.54	−0.0068	NA
	H460		0.72	0.14	0.036	0.0062	2.4	4.8	2.0	0.24	0.051	4.7
	***p value***		***0.005***	***0.012***	***0.087***	***0.14***	***0.017***	***0.0660***	***0.0076***	***0.0048***	***0.049***	***0.0016***

MID, mean inactivation dose.

**Table 2 ijms-20-03635-t002:** Correlation of clonogenic survival parameters between datasets A and B.

Parameters	*R* Values	*p* Values
SF_2_	0.73	0.003
SF_4_	0.81	<0.001
SF_6_	0.86	<0.001
SF_8_	0.91	<0.001
MID	0.87	<0.001
D_10_	0.90	<0.001
D_50_	0.88	<0.001
α	0.79	<0.001
β	0.63	0.014
α/β	0.62	0.037

**Table 3 ijms-20-03635-t003:** *EGFR* mutation status in non-small cell lung carcinoma cell lines.

Cell Line	Histopathology	*EGFR* Status	Reference
H1650	Adenocarcinoma	ΔE746_A750	[23,24]
Ma-24	Adenocarcinoma	L858R, E709G	[23,25]
PC9	Adenocarcinoma	ΔE746_A750	[23,26]
HCC827	Adenocarcinoma	ΔE746_A750	[24,27]
H1975	Adenocarcinoma	L858R, T790M	[24,26,27]
II-18	Adenocarcinoma	L858R	[23,26,28]
H1299	Large cell carcinoma	Wild-type	[23]
H1703	Adenocarcinoma	Wild-type	[23,24]
A549	Adenocarcinoma	Wild-type	[23,24,27,28]
H157	Squamous cell carcinoma	Wild-type	[23,24,28]
LK2	Squamous cell carcinoma	Wild-type	[23,26]
H522	Adenocarcinoma	Wild-type	[23]
H520	Squamous cell carcinoma	Wild-type	[23,24,27]
A427	Adenocarcinoma	Wild-type	[23,28]
H460	Large cell carcinoma	Wild-type	[23,24,27,28]

**Table 4 ijms-20-03635-t004:** Plating efficiency (PE) and the number of seeded cells.

Cell Line	PE (%)	0 Gy	2 Gy	4 Gy	6 Gy	8 Gy
H1650	12	500	500	1000	2000	3000
Ma-24	24	500	500	1000	2000	3000
PC9	25	500	500	1000	2000	3000
HCC827	39	300	300	500	1000	2000
H1975	35	300	300	500	1000	2000
II-18	21	500	500	1000	2000	3000
H1299	71	200	200	300	500	500
H1703	34	300	300	500	1000	2000
A549	72	200	200	300	500	500
H157	65	200	200	300	500	500
LK2	40	300	300	500	1000	2000
H522	43	300	300	500	1000	2000
H520	28	500	500	1000	2000	3000
A427	32	300	300	500	1000	2000
H460	58	300	300	500	1000	2000

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
