# Peer review of "Radiosensitivity Differences between EGFR Mutant and Wild-Type Lung Cancer Cells are Larger at Lower Doses"

_ijms, 2019, doi:10.3390/ijms20153635_

Round 1
Reviewer 1 Report
Dear editor and authors,
the raised points were answered in a satisfactory manner.
Author Response
Reviewer #1:
Dear editor and authors, the raised points were answered in a satisfactory manner.
Response:
We sincerely thank the reviewer for evaluating our revised manuscript.

Reviewer 2 Report
The authors tested the radiosensitivity of a series of wild type- and EGFR mutated non-small cell lung carcinoma cell lines, and analyzed the clonogenic cell survival data on various radiosensitivity parameters. Experimental data are presented from two independent researchers A and B and the obtained datasets A and B show good correlation. Although interesting, this reviewer has a number of critical points regarding the presentation of the data and the conclusions. Questions are specified below.
Major comments:
The authors state, even in the title of their ms., that their data show that radiation response / sensitivity of the tested cell lines, quantified by the different parameters, in particular the SF2, is suitable to identify their genetic profile, i.e. their EGFR mutational status of NSCLC cell lines (e.g. ms. lines 40-41; lines 47-48; lines 69-70; lines 150-151) . The more radiation sensitive cell lines (lowest SF2) show EGFR mutations, and might therefore “best suited for preclinical research aimed at establishing precision medicine biomarkers for radiotherapy” (lines 24-25).
This reviewer completely disagrees with the authors’ conclusion. It is the other way around: the authors demonstrate that EGFR mutated cell lines might be – as analyzed from their clonogenic cell survival curve obtained radiosensitivity parameters – more sensitive to low dose irradiation. The conclusion is therefore, that the genetic profile of NSCLC cell lines might be predictive for their radiation response: cell lines with EGFR mutations might be more sensitive to low doses, low fraction sizes, of radiation. The consequence is that the ms., including the title, requires thorough revision.
A second major point is the presentation of the data in figures 2 and 3. Because, for all cell lines, you tested the full dose range (0, 2, 4, 6 and 8 Gy), this reviewer would like to see the whole cell survival curves, which is much more clear than pooling the SF data what you did in figures 1&2. The parameter data could then presented in a separate table. The great advantage of showing cell survival data:
Results of each individual cell line can be seen, incl. spread of the data (error bars)
Bending of the cell survival curve, which is probably larger for EGFR mutated cells, can be visualized (i.e. visualizing the larger alpha and higher alpha/beta ratio).
Alternatively, the authors could use a different symbol for each cell line that was used, which can be added to the list of cell lines in table 1. Then, in figures 1 & 2, blue and red data points can be replaced by a cell line specific data point. The advantage is again, that the information will be more clear to the readers of the journal. E.g., are all upper bleu dots from the same, probably the most resistant, cell line? Or is there a spread of the data?
To my sincere knowledge, the conclusion from SF2 studies that were performed in the far past, was that for large groups ,a large number of data points, the parameter indeed shows differences in sensitivity between groups of different tumour types. E.g, the mean SF2 of patient derived glioblastoma cell lines is larger than that of lymphoma’s . However, the data show large variation, with the conclusion that the SF2 is not a robust parameter, not suitable, to distinguish between good and bad responding patients upon irradiation. For that reason, the test is not used as a predictive parameter for the radiation response, not one radiation oncologist will change his treatment scheme on basis of this parameter.
Minor comments:
What was the plating efficiency (PE) of the cell lines you tested? This question because of the number of cells you plated (table 3) for the different cell lines, which is probably based on the PE. My suggestion is to add the PE to your table 2 or 3 for completeness.
Lung cancer patients are nowadays often treated with hypofractionation schemes. Your data however show that, for EGFR mutated lung tumours, smaller fraction sizes might be preferential. Or, a better conclusion might be that high fractions sizes in hypofractionation protocols are preferential, because there is no difference in response to be expected between wild type and EGFR mutated tumours. Please comment and consider for the discussion of your ms.
Furthermore, apart from EGFR mutation, the tested cell lines might have more genetic aberrations that influence their radiosensitivity. This should be discussed.
Lines 129- 136: On basis of analyses mentioned in lines 126- 134, I do not understand your statement lines 135-136 saying: “the finding that the SF2 is the clonogenic survival parameter that has the highest ability to predict EGFR status may be generalized regardless of radiosensitivity of cell lines” while you mention that the p-values for the SF2 of radiosensitive cell line comparisons were much lower than for the radioresistant cell lines. Hence , dependent on their radiation sensitivity. Please explain.
Author Response
Reviewer #2:
The authors tested the radiosensitivity of a series of wild type- and EGFR mutated non-small cell lung carcinoma cell lines and analyzed the clonogenic cell survival data on various radiosensitivity parameters. Experimental data are presented from two independent researchers A and B and the obtained datasets A and B show good correlation. Although interesting, this reviewer has a number of critical points regarding the presentation of the data and the conclusions. Questions are specified below.
Response:
We sincerely thank the reviewer for evaluating our manuscript and for providing a number of insightful comments. The manuscript was thoroughly revised according to the reviewer's suggestions as follows.
Major comments:
The authors state, even in the title of their manuscript, that their data show that radiation response/sensitivity of the tested cell lines, quantified by the different parameters, in particular the SF2, is suitable to identify their genetic profile, i.e. their EGFR mutational status of NSCLC cell lines (e.g. ms. lines 40-41; lines 47-48; lines 69-70; lines 150-151). The more radiation sensitive cell lines (lowest SF2) show EGFR mutations, and might therefore “best suited for preclinical research aimed at establishing precision medicine biomarkers for radiotherapy” (lines 24-25). This reviewer completely disagrees with the authors’ conclusion. It is the other way around: the authors demonstrate that EGFR mutated cell lines might be – as analyzed from their clonogenic cell survival curve obtained radiosensitivity parameters – more sensitive to low dose irradiation. The conclusion is therefore, that the genetic profile of NSCLC cell lines might be predictive for their radiation response: cell lines with EGFR mutations might be more sensitive to low doses, low fraction sizes, of radiation. The consequence is that the ms., including the title, requires thorough revision.
Response:
We sincerely thank the reviewer for the insightful and suggestive comments. According to the reviewer's suggestions, the entire manuscript including the title, abstract, and conclusion, was thoroughly revised. The description "predictive ability of clonogenic survival parameters for genetic profiles" was completely deleted throughout the manuscript. Instead, the conclusion suggested by the reviewer "genetic profile of NSCLC cell lines might be predictive for their radiation response; cell lines with EGFR mutations might be more sensitive to low dose-, and low fraction sized-irradiation" was added.
The following parts were revised:
1. the parts pointed out by the reviewer: lines 2-4, 25-27, 42-43, 48-50, 78-79, and 149-151.
2. the parts not pointed out by the reviewer: lines 17-19, 23-24, 52, 106-108, and 143-144.
A second major point is the presentation of the data in figures 2 and 3. Because, for all cell lines, you tested the full dose range (0, 2, 4, 6 and 8 Gy), this reviewer would like to see the whole cell survival curves, which is much more clear than pooling the SF data what you did in figures 1&2. The parameter data could then be presented in a separate table. The great advantage of showing cell survival data: Results of each individual cell line can be seen, incl. spread of the data (error bars); Bending of the cell survival curve, which is probably larger for EGFR mutated cells, can be visualized (i.e. visualizing the larger alpha and higher alpha/beta ratio). Alternatively, the authors could use a different symbol for each cell line that was used, which can be added to the list of cell lines in table 1. Then, in figures 1 & 2, blue and red data points can be replaced by a cell line specific data point. The advantage is again, that the information will be more clear to the readers of the journal. E.g., are all upper bleu dots from the same, probably the most resistant, cell line? Or is there a spread of the data?
Response:
We thank the reviewer for the constructive comments. According to the suggestion, whole cell survival curves are provided as Figure 1 and 2, and the parameter data are presented as Table 1.
To my sincere knowledge, the conclusion from SF2 studies that were performed in the far past, was that for large groups, a large number of data points, the parameter indeed shows differences in sensitivity between groups of different tumour types. E.g, the mean SF2 of patient derived glioblastoma cell lines is larger than that of lymphoma’s. However, the data show large variation, with the conclusion that the SF2 is not a robust parameter, not suitable, to distinguish between good and bad responding patients upon irradiation. For that reason, the test is not used as a predictive parameter for the radiation response, not one radiation oncologist will change his treatment scheme on basis of this parameter.
Response:
We think this is a very important point. We agree with the reviewer's comment that SF2 should not be used as clinical test to predict responses to radiotherapy. On the other hand, evidence suggests clinical applicability of genetic profile-based algorithms generated from correlation analyses of in vitro SF2 data and the genetic profile counterparts (e.g., Strom et al. 2017 Eur J Cancer; Torres-Roca et al. 2015 Int J Radiat Oncol Biol Phys; Strom et al. 2015 Radiother Oncol; Ahmed et al. 2015 Oncotarget). From this perspective, SF2 can be used for pre-clinical study aiming at establishment of genetic profiles that predict tumor radioresponsiveness. This discussion was added in lines 131-138. We thank the reviewer for the comment that improved the quality of our manuscript.
Minor comments:
What was the plating efficiency (PE) of the cell lines you tested? This question because of the number of cells you plated (table 3) for the different cell lines, which is probably based on the PE. My suggestion is to add the PE to your table 2 or 3 for completeness.
Response:
Yes, as the reviewer speculated, the differences in the number of seeded cells among different cell lines are based on the plating efficiency (PE). According to the reviewer's suggestion, the PE data are added in Table 4 in the revised manuscript. We thank the reviewer for the helpful comment.
Lung cancer patients are nowadays often treated with hypofractionation schemes. Your data however show that, for EGFR mutated lung tumours, smaller fraction sizes might be preferential. Or, a better conclusion might be that high fractions sizes in hypofractionation protocols are preferential, because there is no difference in response to be expected between wild type and EGFR mutated tumours. Please comment and consider for the discussion of your ms.
Response:
We definitely agree with the reviewer's opinion. According to the reviewer's suggestion, the following sentences were added in Discussion (lines 127-X130); "NSCLC patients are nowadays often treated with hypofractionation schemes. Our data showing no radiosensitivity differences between EGFR mutant and wild-type groups at high doses (i.e., SF8 and SF6) suggest that hypofractionated irradiation employing high fraction size is likely to achieve comparable tumor control regardless of EGFR status". We thank the reviewer for the insightful comment.
Furthermore, apart from EGFR mutation, the tested cell lines might have more genetic aberrations that influence their radiosensitivity. This should be discussed.
Response:
We definitely agree with the reviewer's opinion. According to the reviewer's suggestion, the following sentences were added in Discussion (lines 145-148). "Realistically, radiosensitivity of the tested cell lines must be influenced not only by EGFR but also by other genetic aberrations; therefore, clinical tumor radioresponsiveness will be more accurately predicted by genetic profiles comprising multiple genes, that warrants further investigation". We thank the reviewer for the insightful comment.
Lines 129- 136: On basis of analyses mentioned in lines 126- 134, I do not understand your statement lines 135-136 saying: “the finding that the SF2 is the clonogenic survival parameter that has the highest ability to predict EGFR status may be generalized regardless of radiosensitivity of cell lines” while you mention that the p-values for the SF2 of radiosensitive cell line comparisons were much lower than for the radioresistant cell lines. Hence, dependent on their radiation sensitivity. Please explain.
Response:
We apologize for the confusion. As is evident from the description "data not shown", these data are preliminary, and therefore, misleading. We now realized that we cannot draw solid conclusion on this matter. To avoid confusion, we deleted the entire section (i.e., lines 129-137 in the original manuscript). We believe that this change will not affect the core value of this manuscript. We sincerely thank the reviewer for pointing out the ambiguous description.
Round 2
Reviewer 2 Report
The authors conveniently answered my questions, critics and remarks, and accordingly revised their manuscript. They changed the title of the ms. and included a part about hypofractionated radiotherapy in their discussion. Depite, this reviewer has some doubt regarding the robustness of the presented clonogenic cell survival data and analysis.
This manuscript is a resubmission of an earlier submission. The following is a list of the peer review reports and author responses from that submission.
Round 1
Reviewer 1 Report
Dear editor,
in the manuscript „Predictive ability of clonogenic survival parameters for genetic profiles associated with tumor radioresponsiveness” the authors determined survival parameters of 15 lung cancer cell lines differing in their EGFR status. Form their results they conclude that the parameter SF2 is a suitable radiosensitivity biomarker for preclinical research.
All together the gain of knowledge by this manuscript is very limited. In my view the suitability of SF2 is well demonstrated in many publications over decades and the clonogenic assay has a high tradition in radiation biology and is accepted as gold standard. The authors themselves stated this in the discussion (see citation 8-11).
- Data showed that cell lines with mutated and wildtype EGFR can be separated by SF2, not more. Therefore the generalization in the title is not appropriate. Where is the connection to genetic profiles? EGFR is only a model system.
- The biological consequences of the individual mutations should be validated, is there a correlation between type of mutation and SF2?
- Radiosensitivities in Set A and B cover a large spectrum, what is the correlation for the individual cell lines between set A and B?
- Description in methods is incomplete, important parameters like number of seeded cells is missing
- What is about the robustness of the clonogenic survival assay? Can variable cell numbers be seeded?
- SF2 is claimed as most appropriate parameter, can this be generalized or does it depend on the radiosensitivity of the cell line?
- in a recent publication (https://doi.org/10.18632/oncotarget.24448) the same authors suggest D10 as suitable parameter to evaluate Clonogenic survival, it should be discussed why this is not the case in this study.
- An exemplary presentation of a survival curve including the analyzed parameters would be helpful for non-experts